# Mapping the Sea on Scotland's Peripheries

**Inge Panneels** 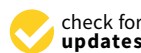

Creative Informatics, School of Computing, Edinburgh Napier University, Edinburgh EH11 4DY, UK;
i.panneels@napier.ac.uk

**Abstract:** This paper examines the use of mapping methodologies in some recent examples of contemporary art that chart the layered seascapes of the remote coastlines on North West Scotland as seen through the lens of visual culture in the Anthropocene. The art projects interrogate conflicting perspectives on landscape and nature in the North. The case studies demonstrate, both directly and indirectly, the political and cultural tensions made evident by the mapping of the micro and macro undercurrents at work in the region, and examine how mapping has been used as a methodology to visualise those intractable material relationships, often using the map as a trope to do so. These mappings make visible the enmeshments of these remote locations into a global ecosystem. The concept of the Anthropocene provides a useful framework to describe the contemporary context of climate change, ecological decline, biodiversity loss and recent discourses on land use within which the artworks by two artists, Julia Barton and Stephen Hurrel, will be discussed. The significance of Kester's concept of Littoral Art were explored through the eponymous art project by Barton, which maps the human debris brought by the northern sea currents to the shores of the Western and Northern coasts, and Stephen Hurrel's cultural mapping of the island of Barra on the West Coast. These projects were further considered in the context of Timothy Morton and Tim Ingold's meshwork theory and the concept of the 19th century Scottish town planner and environmentalist Patrick Geddes, whose urging to 'think global, act local' is implicit in the multi-layered understanding of the Anthropocene.

**Keywords:** mapping; cultural ecosystems; peripheries; the North; Scotland; the sea; Anthropocene; creative geographies; critical cartographies; experimental geographies

## 1. Introduction

The current climate change emergency demands a deeper understanding of the entanglement of climate and culture as a prerequisite to climate action. This article examines the ways in which recent artists' projects in Scotland have employed the map and mapping to chart environmental changes. It makes the link between these mapping practices as legacy of the 20th century avant-garde artist and ecological pioneer Joseph Beuys' radical concept of Social Sculpture and the concept of 'think global, act local', attributed to the 19th century Scottish biologist, sociologist and town-planner Patrick Geddes (Stephen et al. 2004). Patrick Geddes was not only a planner and ecologist but also an art activist and a pioneer of community-based town planning who argued for the symbiotic sustaining of both the environment and culture, which makes his legacy particularly relevant in relation to the activist mapping discussed here.

For Geddes, it was the 'two intertwined strands of the regeneration and the sustaining of the environment on the one hand, and of the revival and sustaining of culture on the other' which were fundamental to all his thinking (Macdonald 2004, p. 61). Geddes saw both environmental and cultural thinking as interdependent, and he explored this fundamental tenet through a multi-disciplinary practice of which synergy was a key foundation.

The understanding of how maps *work*, and the diverse ways in which *space* and *place* are conceptualised and analytically employed to make sense of the world has been analysed more in the last three decades than ever before. David Harley (1989) contended that the process of mapping consists of creating, rather than simply revealing, knowledge, which opened up a new discourse of critical cartography within the discipline of geography. This is a critical aspect of mapping that is examined in the case studies. Denis Wood argued in *The Power of Maps* (1992) that understanding the historical provenance and the rationale of maps to claim territories, mark property and denote political boundaries, makes maps potent tools of power. If maps 'always lie', and maps are never complete or true, as the one paragraph story from the *Exactitude of Science* by Jorge Louis Borges (1946) has famously demonstrated, then maps are transitory, fleeting, relational and context dependent.[1] Wood (1992), together with Mark Monmonier (1991), Kitchin and Dodge (2007) and John Pickles (2004), extensively interrogated 'the power/knowledge relationship of maps' (Kitchin and Dodge 2007, p. 332).

The democratisation of mapping—or the 'undisciplining of cartography'—is what Foucault called 'an insurrection of knowledges' (Crampton and Krygier 2006, p. 12). Mapping is a constant process of re-territorialisation: a spatial practice to solve relational problems (Massey 2005; Kitchin and Dodge 2007). This understanding of mapping will help frame the art practices discussed further in the context of Littoral Art (Kester 2000). The recent move away from the map towards mapping as a methodology has coincided with a move towards socially engaged art practices, where, since the 1960s, artistic practices have appropriated social forms as a means to bring art closer to everyday life (Bishop 2004, 2006; Thompson 2012), as argued for by Joseph Beuys' notion of Social Sculpture, of a connective practice towards social and ecological justice. It argued for a model of art that was total—inclusive, participatory and multi-dimnesional. Art, Beuys argued, does not operate outside of life. This article locates these mapping projects specifically within environmental art practices. Critical cartography (Crampton and Krygier 2006), the linking of maps with power and thus an active and political agent, has become a powerful tool for socially engaged arts practices. Critical cartography in the hands of particular artists has become radical cartography (Bhagat and Mogel 2008), responsive to a political moment, temporal and anti-monumental as a form of resistance, whilst at the same time, it has become an experimental cartography (Thompson and Independent Curators International 2008) where mapping may be used to visualise and imagine radically new ideas of world making. Critically, experimental geography as Trevor Paglen argued, 'expands [Walter] Benjamin's call for cultural workers to move beyond 'critique' as an end it itself and to take up a position within the politics of lived experience (Paglen in Thompson and Independent Curators International 2008, p. 32). This move into the 'practice' he argued, creates new spaces and new ways of being.

The concept of the Anthropocene (Crutzen and Stoermer 2000) provides a useful framework to describe the contemporary context of climate change (IPPCC 2018), ecological decline and biodiversity loss (Kolbert 2014) that is attributed to the impact of human activities on other ecosystems. The understanding of interdependency of human and non-human and local and global ecosystems has been quantified in the Ecosystem Services approach as commissioned by the U.N. Millennium Ecosystem Assessment (2005), which led to the UK National Ecosystem Assessment (2011) and Follow-Up (2014), with a special report that focused on Art and Humanities Perspective on Cultural Ecosystems Services (Church et al. 2014; Coates 2014) which found that Cultural Ecosystems are integral to the overall Ecosystem. In this sense, the Ecosystems Services embody the ethos of Geddes.

---

1  From the *Exactitude of Science*, by Jorge Louis Borges (1946): ' . . . In that Empire, the Art of Cartography attained such Perfection that the map of a single Province occupied the entirety of a City, and the map of the Empire, the entirety of a Province. In time, those Unconscionable Maps no longer satisfied, and the Cartographers Guilds struck a Map of the Empire whose size was that of the Empire, and which coincided point for point with it. The following Generations, who were not so fond of the Study of Cartography as their Forebears had been, saw that that vast map was Useless, and not without some Pitilessness was it, that they delivered it up to the Inclemencies of Sun and Winters. In the Deserts of the West, still today, there are Tattered Ruins of that Map, inhabited by Animals and Beggars; in all the Land there is no other Relic of the Disciplines of Geography'. Purportedly from Suárez Miranda, Travels of Prudent Men, Book Four, Ch. XLV, Lérida, 1658.

As mapping emerged as a distinctive critical tool for visual artists, as evidenced in a range of practices in recent decades, defined as *critical cartography*, *radical cartography* or *experimental geography*, these mapping practices have a concomitant focus on action and activism and define certain collaborative or socially engaged practices. These are 21st century cultural mapping practices, as understood in the context of Cultural Ecosystems Services, a framework that outlines the enmeshment of culture and nature and demonstrates the non-material benefits of nature. The experimental geographies explored in the mapping practices of the artists discussed in this article are spatial practices of culture and evidence of cultural ecosystem services.

## 2. Littoral: Neo Terra

> The sea is a body in a thousand ways that don't add up, because adding is too stable a transaction for that flux, but the waves come in in a roar and then ebb, almost silent but for the faint suck of sand and snap of bubbles, over and over, a heartbeat rhythm, the sea always this body turned inside out and opened to the sky, the body always a sea folded in on itself, a nautical chart folded into a paper cup.
>
> —Rebecca Solnit (2007, p. 255)

For Rebecca Solnit, the seashore is an edge, 'perhaps the only true edge in the world where borders are otherwise mostly political fictions' (Solnit 2007, p. 254). The seashore is perpetually in flux, defined by a travelling body of water. This littoral edge forms the focus of Julia Barton's *Neo Terra* (2016) exhibition that interrogates the Anthropocene through her engagement with coastal marine plastic pollution on the North-West coast of Scotland: from Ross and Cromarty to the North of Shetland. These remote beaches, synonymous with an image of Scotland as a tourism destination, are not fundamentally remote but intrinsically connected to the global ocean systems. By examining the impact of plastic waste in the Anthropocene, Julia Barton's *Neo Terra* (2016) invites 'visitors to take a new look at the scale, nature and consequences of this environmental issue'.[2] Two artworks from this exhibition, *Archipelago and Sample*, are discussed as encountered during the author's field trip to the *Neo Terra* exhibition at An Talla Solois, in Ullapool in June 2017 (Figure 1).[3]

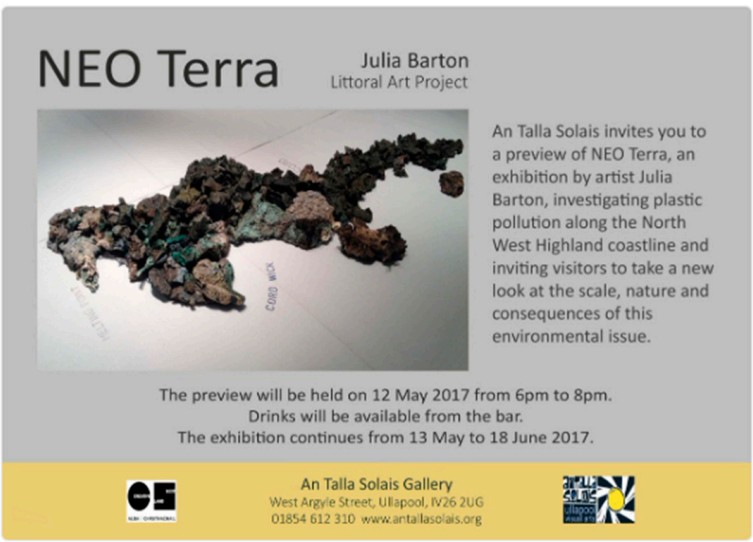

**Figure 1.** *Neo Terra* exhibition invite. An Talla Solais, Ullapool. 13 May–18 June 2017. Photograph created by the author.

---

[2] Wording on the Preview Invitation.

[3] *Neo Terra* (2016) was conceived for the eponymous exhibition at the Shetland Museum and Archives in Lerwick (8 October–12 November 2016) and was later exhibited at An Talla Solais Gallery in Ullapool (13 May until 18 June 2017).

Barton instigated the ongoing *Littoral: sci-art* project in 2013 after walking on a beach on the North-West coast of Scotland 'so thick with mounds of litter that I feared drowning in it' (Barton 2018). Its title, not just a pun on the word litter, refers to the zone of the sea close to the shore where most of the debris is washed up. Solnit likened the shore to an open border between the known and the unknown, a meeting ground and her actions of 'wandering the coastline with downcast eyes to find what there is to be found, a material correlation to composing and thinking', as 'a disreputable profession with its own word, beachcombing' (Solnit 2007, p. 254). Critically, littoral also refers to the *dialogical aesthetics* of Littoral Art as defined by Grant Kester as 'discursive aesthetic' where process is as critical to the practice as the artwork itself and which breaks down the conventional distinction between artist, art work and audience (Kester 2000, p. 4). Littoral Art has a distinct interdisciplinary nature that operates between discourses (art and activism for example) and between institutions (gallery and community for example). The curator Nato Thompson (2015) noted that the emergence of particular durational art projects coincided with a move away from the production of material goods in the 20th century towards the ephemeral production of 'meaning' in the 21st. These projects, Thompson observed, are indicative of relational aesthetics (Bourriaud 2002), and it is argued of littoral art or dialogic art, and are often bannered under the rubric of socially engaged art (Bishop 2004; Bishop 2006; Thompson 2012). Here, Littoral Art can be understood in relation to avant-garde artist Joseph Beuys' (1921–1986) concept of *social sculpture*, which 'encourages and explores transdisciplinary creativity and vision towards the shaping of a humane and ecologically viable society' (Social Sculpture Research Unit 2012).

Barton walked many miles with 'downcast eyes' to find washed up debris. She came to recognise its origins and reconstructed its material journey by charting it appearance on a map it and telling its story. Her long-term investigation of marine plastics focused on 'plastiglomerates': lumps of burnt plastic that have been melted and, in the process, bonded with natural beach materials (Figure 2). *Science* magazine noted that recent anthropogenic deposits contain new minerals and rock types, which reflects their rapid global dissemination. These deposits often contain new materials, such as elemental aluminum, concrete and plastics, and are thus referred to as 'technofossils' (Water et al. 2016). Plastic is therefore a critical marker of the Anthropocene. In 'Life and Death in the Anthropocene: a short history of Plastic' (2015), Heather Davis traced the origins of plastic and its links to petrocapitalism argueing that it represents the promise of modernity: 'the promise of sealed, perfected, clean, smooth abundance'.[4] Yet, the plastic encountered by Barton no longer holds that promise: this is plastic ragged, torn, deformed, melted and misshapen and very much enmeshed with nature. Davis observed the 'accidental or incidental aesthetics' of the Anthropocene and noted that the aesthetic effects—'as in aesthesis or affects produced by our sensorial experience of the environment' have been drastically altered by the presence of plastic (Davis and Turpin 2015, pp. 348–49). Even though the substance may break apart, degrade and become microscopic: it does not biodegrade. As Barton noted, it never goes away. Thus, plastic has moulded both the material and visual culture of the Anthropocene.

---

[4]    Plastic represents the 'inevitable corollary' of unfettered economic growth: consuming eight percent of the world oil production, and ecologically devastating with a projected 33 million tons of plastic produced annually by 2050, and a growing understanding of its insidious effects on human and non-human life (Brown et al. 2016) The first synthetic polymer was invented in 1907 by Belgian chemist Leo Bakeland (then referred to as Bakelite), which ultimately led to the development of plastics, which would re-make and re-invent the world as we know it today.

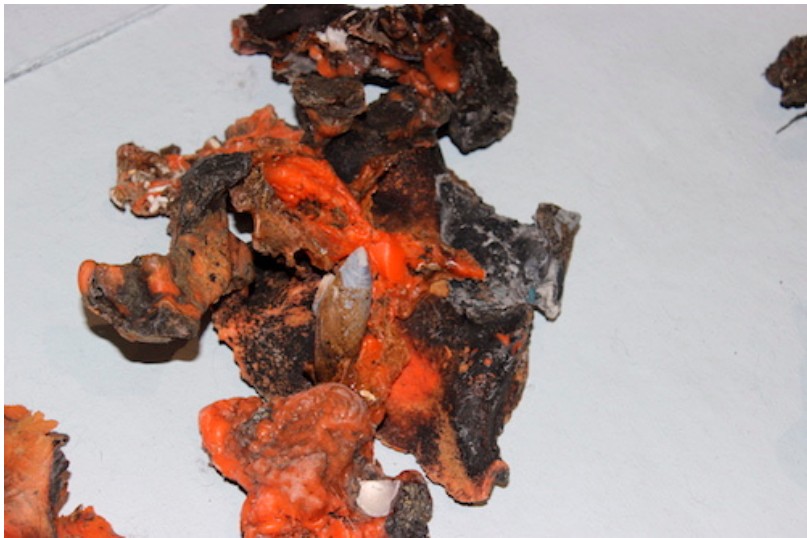

**Figure 2.** Plastiglomerate with embedded mussel shell. *Neo Terra* exhibition, Ullapool (2016). Photograph created by the author.

*Archipelago*, the centrepiece of the *Neo Terra* exhibition, was a largescale floor-based map featuring 'islands' composed of *plastiglomerates* (Figures 3–5). Plastic is burnt, mostly by the fishing industry in Scotland, offshore or onshore, as a means of managing plastic debris. These plastic forms ultimately resemble rock formations, but that appearance belies a very different texture and weight. These lightweight forms were laid out by Barton on a large canvas map with co-ordinates and names written in pencil. The naming of the islands was informed by discussions with school children.

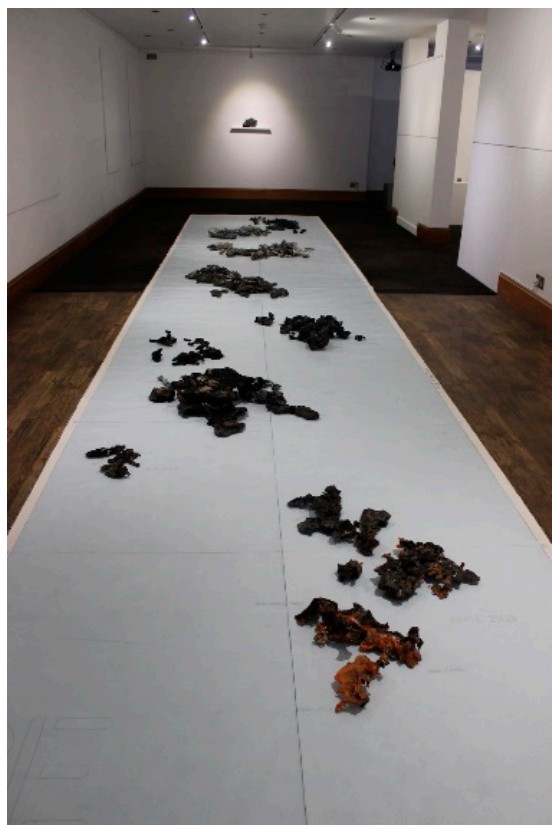

**Figure 3.** *Archipelago*. Floor installation overview and detail.

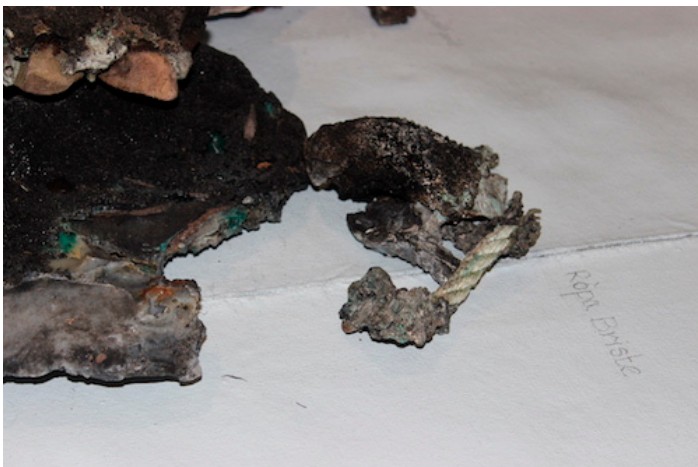

**Figure 4.** *Archipelago*: detail. Photograph created by the author.

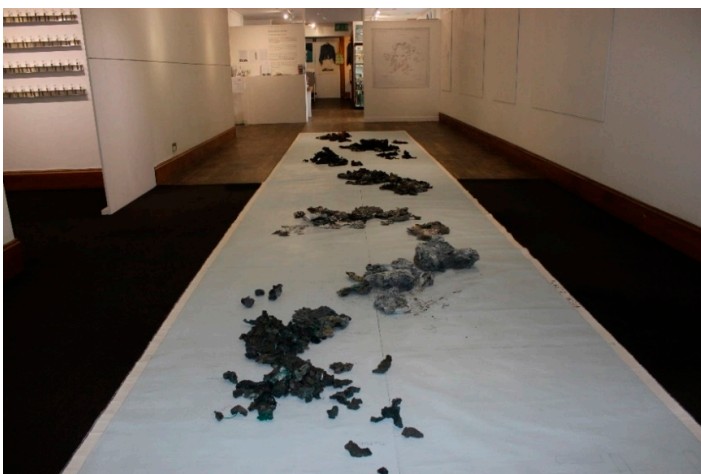

**Figure 5.** *Archipelago*. Exhibition overview. Photograph created by the author.

Barton approached the *Littoral* project in a methodical manner, taking cues from science in her method and display, and immersed herself fully in both the environment and the subject of the project. She followed the 'art of enquiry' as Ingold (2013, pp. 6–8) put it. The lab coat hung on a peg, the microscope, the test tubes and petri dishes on display, embodied a scientific aesthetic. The scientific methodology was evident equally in its scope and rigour. The *Sample Installation* depicted a map of the North West of Scotland drawn in graphite onto the gallery wall, and listed sixty beaches stretching from Garloch on the coastline of Wester Ross to Durness on the northern-most coast of Sutherland. Each beach was plotted on the map, and a sample of the sand of the corresponding beach collected. The resulting sixty samples were displayed in glass test tubes in five rows of twelve, each labelled with a corresponding number of its geographical location (Figure 6). Each sample was subsequently analysed under the microscope and an image taken (Figure 7). Thus, a systematic survey of the Ross-shire coast was being undertaken. Several samples were severely contaminated with plastic micro beads. As Barton noted in the text accompanying the exhibition, microbeads have entered the food chain at an alarming rate. Up to 236,000 tons of the plastic per year breaks down into *microbeads* (less than 5 mm across).[5] These make their way up the ocean food chain into fish to finally enter into human food

---

5　Worldwide, more than 300 million tons of plastic are produced annually, of which 10% ends up in the oceans. It is estimated that there is now a 1:2 of plastic to fish ratio and that plastic will outweigh fish by 2050 if the problem is left unchecked (Munro 2010). When ingested, if these microbeads do not directly kill the animals, the toxic effects of the chemicals may affect their hormones levels and behaviour (Brown et al. 2016).

(Brown et al. 2016). Thus, each sample spoke not only of its location but also of the wider implications of global marine pollution.[6] In an interview with the artist (18 June 2017), Barton mentioned that she had been contacted via social media during the exhibition by a Canadian scientist who was very interested in the collected sand samples which potentially highlight the extent to which sea-sand travels, as each grain of sand carries within it a trace of its source.[7] The extent to which both sand and plastic travels has only begun to be better understood in the last two decades (Tweedie 2018).[8]

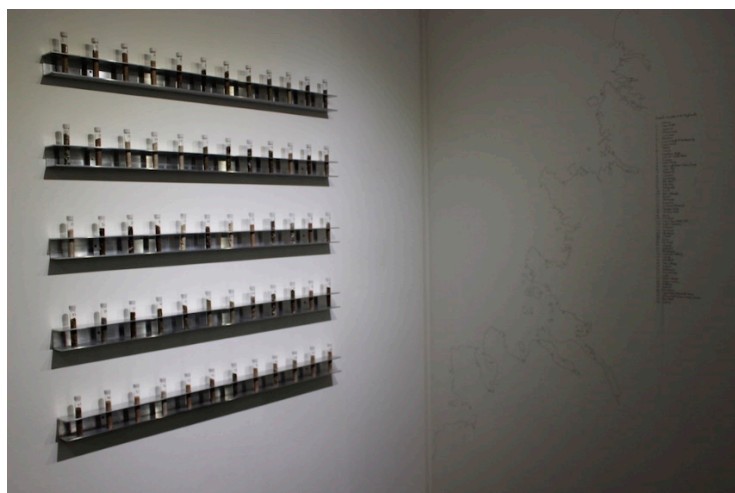

**Figure 6.** Sample, installation at *Neo Terra* exhibition, An Talla Solais, Ullapool. 13 May–18 June 2017. Photograph created by the author.

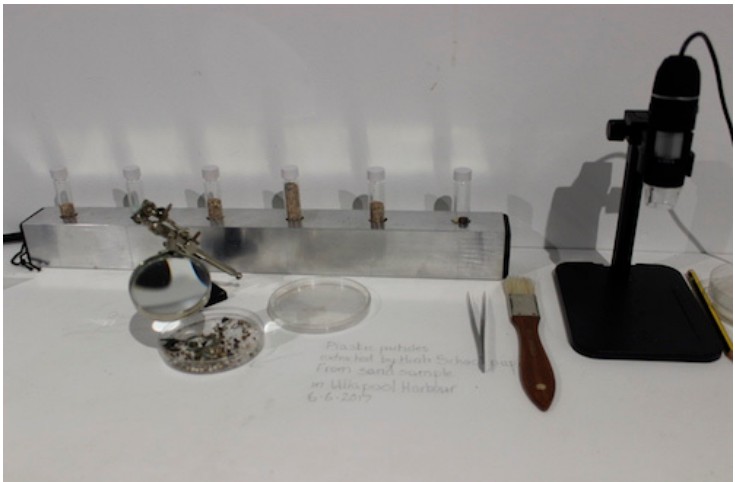

**Figure 7.** Interactive Lab at *Neo Terra* exhibition, An Talla Solais, Ullapool. 13 May–18 June 2017. Photograph created by the author.

---

6    In an experiment conducted by Icelandic scientists in 2016, two plastic containers fitted with GPS trackers were released of the southern Iceland coast to demonstrate where plastic discarded in the ocean ends up. Both floated west, past Greenland and towards Canada before drifting east: one of the samples ended up on the Faroe Islands and the second one on Tiree in Scotland (McKenzie 2017). This 'message in a bottle' thus highlighted the connectedness of ocean currents, as had been demonstrated by a previous accidental experiment in 1992 when 28,800 plastic bath toys lost at sea in the North Pacific were still washing ashore two decades later. The geolocation of the emerging toys helped scientists to map previously unknown global ocean currents (Hohn 2011). It also highlighted the issue of plastic pollution, often ingested by marine wildlife. The North Pacific is also the location of the Great Pacific Garbage Patch; an island of floating debris estimated to be twice the size of the state of Texas.

7    Julia Barton in conversation with author during the exhibition visit at An Talla Solois in Ullapool, on 18 June 2017.

8    Sand has now been identified as another resource at risk of depletion (Tweedie 2018).

As noted above, *Neo Terra* thus connects the plastic washed up on these local remote beaches in the North West of the British Isles to a wider, global issues. The project also invites us to examine these beach samples through the microscope and to better observe the issues at stake. For Barton: 'What you actually see on beaches is just the tip of the iceberg and nothing compared to the unseen particles which break down into sand grains' (in interview 18 June 2017). A public awareness campaign by Greenpeace in 2017 saw a giant beached whale filled with plastic washed up on a beach in the Philippines (Pajada 2017). On closer inspection, the whale itself was made up of plastic rubbish, as part of an installation: art imitating life. It is 'playing for time, making art as if the world mattered', as Lucy Neal noted in the title of her book on activism and art (Neal 2015). It was exactly the activist agenda of *Neo Terra* which found particular resonance with its audience. *Neo Terra* gave visitors the space and time to think about plastic marine pollution on many levels, as reflected in the comments book of the exhibition:

> thought-provoking
>
> a fabulous demonstration of what is affecting our health and environment
>
> gave me a chill up my spine
>
> to create something so thought-provoking and beautiful out of something so shocking is a triumph
>
> outstandingly important
>
> challenging, authentic, beautiful and ultimately extremely horrifying. Art is a powerful medium to highlight this environmental catastrophe. Amazing work
>
> disturbingly beautiful art but so much food for thought
>
> <div align="right">Littoral Art[9]</div>

The comments validate the role of art as not only having an aesthetic function (e.g., 'beautiful') but with an ability to shock, challenge and jolt our perception, compelling us to take up a position towards action. However, the exhibition—and the comments book—belies the deep engagement that underpins this project. Workshops with communities, including school children, in Shetland and Ullapool were involved in the *process* by collecting (litter and sand samples), making and naming of the work (*Archipelago*), the feedback (comments book and talks). A harbour master asked Barton to get in touch as a possible ally in his quest to clean up his harbour, when other projects had failed. Community engagement formed a key part of the *Neo Terra* project and continues to be an essential part of the overall ongoing *Littoral: sci-art* project. Therefore, it was only a logical next step for the *Neo Terra* project to travel to the Scottish Parliament in Edinburgh in December 2017 to highlight the impact of marine plastics to ministers and MSPs and speaks of a deep political engagement. As Thompson and Independent Curators International (2008) had noted, art is a powerful interlocutor, which affects people on a deep, emotional level, but with this comes a profound responsibility. Thus, it is the 'dialogical' relationship of the work in its *discourse* (in interviews/discussion by the artist with actors/agents in the process of plastic pollution as well as public talks), and through the *process* of making that the 'relational' aspect of the work makes this Littoral Art work, operating between discourses of science and art, art and activism.

The Anthropocene has arguably ended the certainty of Modernity, ironically called for by scientists themselves (Latour 2013 in Chandler 2018, p. 5). Modernity has been defined by a belief in progress, a faith in rationality, often with a linear causality and a Cartesian divide between culture and nature. The Anthropocene upends all this (Lewis and Maslin 2018). These qualitative

---

9    You can read the scanned comments online. Available at: https://littoralartproject.com/neo-terra-exhibition-comments/ (accessed on 1 July 2018).

responses, enable an articulation and understanding of the abstract concept that is the Anthropocene. This enmeshment of relationships is presented by Bruno Latour in terms of Actor Network Theory (ANT) (1996) which Latour saw fundamentally not as much a theory as a method (Müller 2017). ANT has become a key inspiration for geographers as a means of incorporating materiality into geographical theory and practice. It is argued that this pre-occupation with all things material, could be seen as a counterbalance to the cultural turn in the late 1980s, which had focused geographers' attention on meaning and representation (Rogoff 2000; Kitchin and Dodge 2007; Kong in: Müller 2017). Latour argued that ANT reverses the background to the foreground: instead of starting from universal laws—social or natural—it starts from 'incommensurable, unconnected localities, which then, at a great price, sometimes end into provisionally commensurable connections' (Latour 1996, p. 4). He argued that the most counterintuitive aspect of ANT is that there is literally nothing but networks, with nothing in between them. He uses the old physics metaphor of the aether: there is none. However, Ingold (2013) differentiates between networks and meshworks. Networks, he argues, are 'a spatial construct' with lines that connect. Meshworks on the other hand, do not connect but are lines of movement, or growth. They are 'temporal lines of becoming' argued Deleuze and Guattari (2004, pp. 224–25). Where networks have nodes, meshwork have knots: its ends always loose 'where it is groping towards an entanglement with other lines, in other knots. What is life, indeed, if not a proliferation of loose ends!' (Ingold 2013, p. 132). This is a concept Timothy Morton also explored (Morton 2007, 2010, 2011). Meshworks, Morton argued, 'can mean the holes in a network, and the threading between them' (2010, p. 28): without the 'holes' that weave the fabric there would be no mesh. For Morton, the mesh equates all living and non-living things. The mesh is not static but is infinite in its connections, in its scale and differences: there is no foreground nor background. Barton's work can be usefully analysed via the concept of meshworks, as they reveal the connections between disparate materials on the move (plastiglomerates, plastic fibres, plastic earbud sticks, etc.), from different geographical locations (sixty beaches in Scotland and Canada), across timescales and bringing together multiple interpretations of meaning (scientist, fisherman, artist, school children, etc.). The plastic tossed or lost at sea is no longer in the background of the big wide open ocean but reveals itself in a meshwork, floating on the surface, enmeshed with life, visibly (Figure 2) and invisibly as microbeads (Figure 7), everywhere, ever returning, always, 'in a thousand ways that don't add up' (Solnit 2007, p. 255). As Latour argued, 'instead of having to choose between the local and the global view, the notion of network allows us to think of global entity—a highly connected one—which remains nevertheless continuously local . . . ' (Latour 1996). However, rather than referring to the art project as revealing a network, the differential of meshwork is more appropriate: Barton's work has the properties of a meshwork, with loose ends, always groping towards other connections and yet connected like the weave in fabric, undulating like waves in the sea. The meshwork concept that embodies the Littoral Art project finds resonance in Patrick Geddes' urging to think global, act local. The mapping practices of *Neo Terra* have exposed a meshwork of mobile connections between economic activity at sea impacting social behaviour off- and onshore (burning of plastic), affecting human and non-human life. The maps visualised those knots of connectivity of the meshwork.

## 3. Mapping the Sea: Barra

Stephen Hurrel also mapped the sea on the West coast of Scotland, but focused on the relational aspects of the sea rather than material enmeshments. Three artworks by Hurrel: *Belonging to the Sea* (2012), *Mapping the Sea* (2013) and *Sea Stories* (2013) discussed here, chart the innate and intimate knowledge of the sea by local communities on the Isle of Barra on the Outer Hebrides. These are part of a larger body of work that interrogates this seafaring culture and touches upon the inherent tension between local, indigenous knowledge and governmental policy making. Hurrel, whose family hail from Barra in the Outer Hebrides, says of his work: 'I make art that grows out of an engagement

with specific contexts. It is a relational art practice that takes into account the wider ecology of people and place'.[10]

Hurrel's last two works were made under the auspices of Cape Farewell's *Sea Change* project, which, much like Barton's Littoral project, was a long durational art project that took place over four years across Scotland's western and northern isles.[11] The *Sea Change (Tionndadh na Mara)* project began in 2010, with a gathering of artists and scientists from across the UK at artist residency centre Cove Park on the West Coast of Scotland and was conceived by Ruth Little.[12] In 2011, thirty UK and international artists and scientists sailed on a Marine Conservation research vessel *Song of the Whale* between the Hebrides (Mull, Small Isles, Skye, Lewis, Rona and the Shiants) and St Kilda. The second expedition on board the *Swan* in 2013 took place in Orkney and Shetland. Artists and scientists worked collaboratively, or independently, to consider 'the relationships between people, place and resources in the context of climate change'.[13] The artists, together with the scientists, worked with local mainland and island communities to produce workshops and present work-in-progress.[14] The resulting *Sea Change* exhibition took place at the Royal Botanical Gardens in Edinburgh from 8 November 2013 to 26 January 2014 and brought together the 28 artists who took part in the two expeditions. The title, *Sea Change*, borrowed from Shakespeare *The Tempest* (1610), was an aptly chosen metaphor for the radical change or transformation the project sought to interrogate.

> Full fathom five thy father lies:
>
> Of his bones are coral made;
>
> Those are pearls that were his eyes:
>
> Nothing of him doth fade
>
> But both doth suffer a sea-change
>
> Into something rich and strange.
>
> Sea nymphs hourly ring his knell
>
> ——Shakespeare, *The Tempest*, 1610

---

[10] Hurrel grew up on the West Coast and is influenced by its landscape. He witnessed the submarines going past: thus, the dichotomy of the manmade and the natural environment are often explored in his work. *Beneath and Beyond* (2008) for example, and shown at Tramway, Glasgow, made the live sound of the earth visible. The invisible nature of what is happening beneath your feet—the sound of low frequency seismic movement across the world—is made tangible (in interview: 1 February 2019).

[11] The *Sea Change* project formed part of London 2012 Festival, the Year of Creative Scotland (2012) as well as the Year of Natural Scotland (2013) and was supported by Arts Council England, Creative Scotland, Cove Park and The Bromley Trust.

[12] Cape Farewell was set up in in 2001 by artist David Buckland following a collaboration with climate modellers at the Met Office Hadley Centre for Climate Change and Sciences. Buckland observed the capability of climate data to project into the future. He argued that humanity never had had such as tool and that in the past, 'it was the role of artists and visionaries to map futures, but with no sense of logic or probability'. It is thus through the modelling and mapping that the future has become visible. Cape Farewell developed a model of recrafting abstract science data into more urgent narratives informed by expeditions and action-based research. Notably, in 2007, Cape Farewell changed tact and acknowledged that climate change was a given scientific fact and instead focused on solution-based projects. Cape Farewell continue their current cultural activity 'towards building a Renaissance in energy production, economic regeneration and a cultural renewal'.

[13] Project details taken from Cape Farewell's website. Available from: http://www.capefarewell.com/latest/projects/sea-change.html (accessed on 7 July 2017).

[14] Cape Farewell has an international remit and ambition: since 2003 it has led eight expeditions to the Arctic, two to the Scottish Islands, and one to the Peruvian Andes, taking artists, scientists, educators and communicators to experience the effects of climate change first-hand: 'By physically sailing to the heart of the debate, Cape Farewell aims to draw people's attention to the effects of ocean currents on us and our climate'. Sustained critique on Cape Farewell has noted the inherent contradiction in the large carbon footprint of these expeditions and their inherent elitism by selection of (far off) sites and 'A-listed artists' (Smith and Howe 2015). Cape Farewell has diversified their programme—possibly in response to this criticism- by tracking climate change in less remote locations and with a focus on local history and place. The Scottish expeditions are, in short, a response to this criticism.

The *Sea Change* expeditions explored island communities in Scotland and 'their work to make themselves self-sustaining and carbon neutral'. As such, they invited artists who were from or lived and worked on the islands: Barra, Eigg, Lewis, Canna, Orkney. With his long-term investigation of socio-ecological issues of marine-based environments, Hurrel's participation in the Cape Farewell expedition of 2011 was valuable and introduced him to the work of social ecologist Ruth Brennan who worked for the Scottish Association for Marine Sciences, based in Oban. They continued to work collaboratively on a number of projects: *Belonging to the Sea* (2012), *Sea Stories* (2013), *Mapping the Sea: Barra* (2013) and *Clyde Reflections* (2014). This paper discusses the Littoral Art practice of Stephen Hurrel by analysing the first three collaborative art works, which were all sited in Barra.

In an interview (1 February 2019), Hurrel commented that the geographical boundaries of these small island communities made them easier to define and interrogate. He also observed that communities such as Eigg are at the forefront of sustainable living experiments who have since the island buyout managed to increase the population yet decrease the island's carbon footprint by becoming self-sustainable in energy terms.[15] As Hurrel noted (interview 1 February 2019), his interest in landscape and ecology, was informing his work which was exhibited and made in urban landscapes. However, he felt that something was 'missing' and the opportunity to go back 'into the landscape' proved pivotal. The commissioning by Cape Farewell of *Mapping The Sea: Barra* was thus the culmination of a prolonged period of extensive mapping of the sea there, in multiple iterations (*Sea Stories, Belonging to the Sea* and *Mapping the Sea: Barra*), explored by interdisciplinary approaches and understanding of *site* by Hurrel and Brennan. The map was used as a tool to elicit data, to collate and make visible a cultural indigenous heritage, mapping stories across time and space that will enable future generations to understand the sea from more perspectives than digital mapping technologies alone would allow.

*Dùthchas na Mara/Dúchas na Mara/Belonging to the Sea* (2012) was an existing project by Ruth Brennan and Iain MacKinnon (Scottish Crofting Federation) and Barra fishermen. Hurrel made the cover map and took the photographs and video footage (Figures 8 and 9). This project collected stories and explored the roots of maritime conflict over fisheries and marine conservation on Gaelic-speaking islands in Scotland and Ireland and was published in English, Irish and Scottish Gaelic (MacKinnon and Brennan 2012).[16] Hurrel described how one of the project aims was to look at and describe these remote communities who were living on subsistence farming only a generation ago and ask what strategies they deploy to survive. In the interview (1 February 2019), Hurrel described a resilient community structure, which, he argued was, in the past, provided for by belief systems: religion, morals, traditions, etc.

---

[15] The island of Eigg was bought for £1.5 million by the local community in 1997 with support from the public and the Scottish Wildlife Trust. It has since made concerted effort to derive all the island's energy needs from sustainable sources: wind, water and solar, winning several awards for its efforts (Scottish Wildlife Trust 2017).

[16] Translations by Rody Gorman. It was published in a free full-colour booklet. and was distributed to local people at the Barra Community Hall on 15 August 2012. It is also available online. Available at: http://www.sams.ac.uk/ruth-brennan/belonging-to-the-sea/view (accessed on 8 March 2016). The project received a First in a Lifetime Award from Creative Scotland.

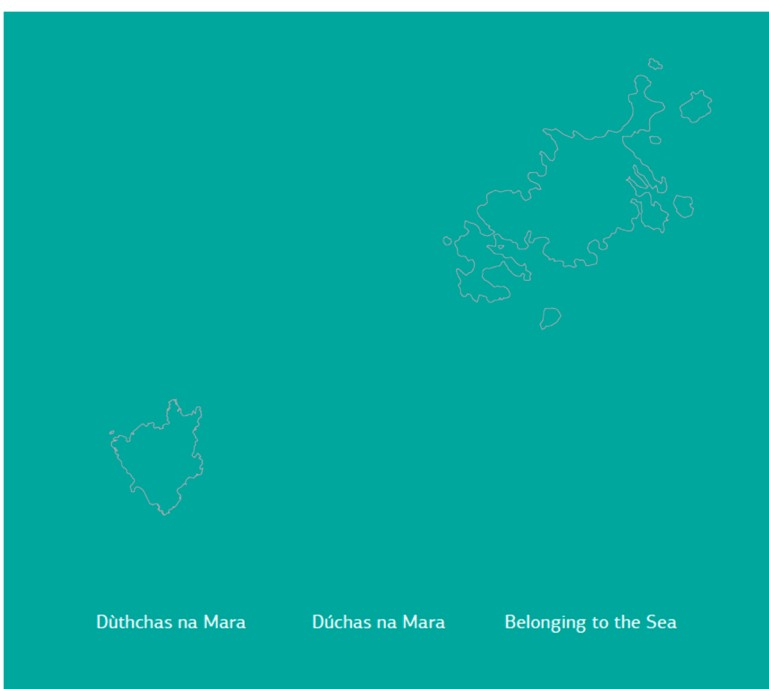

**Figure 8.** Belonging to the Sea (2012) cover map by Stephen Hurrel. e-book. Image used with permission form the artist.

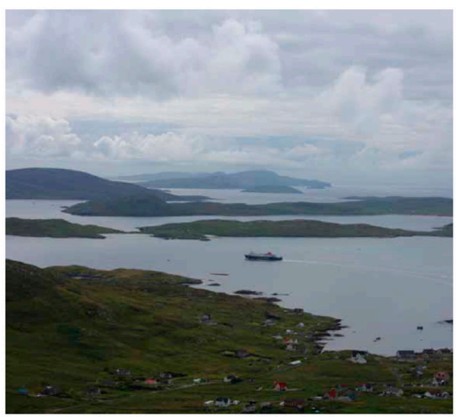 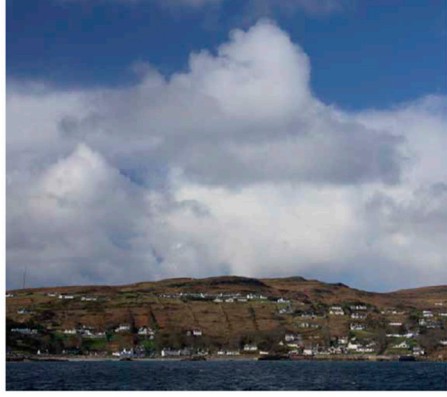

**Figure 9.** Photographs by Stephen Hurrel, Belonging to the Sea, double spread page, p4 e-book. Image used with permission form the artist.

The traditional resilience of the Gaelic community in living on the periphery, 'of weathering the storm' is considered very relevant today in consideration of sustainable ways of living. This point was made by Joanne Orr, Chief Executive of Museums Galleries Scotland (MGS), arguing the case for the inclusion of the UK in the Intangible Cultural Heritage (ICH) Convention 2003 as part of global sustainable development strategies and cultures (UNESCO 2012).[17]

The researchers showed maps and admiralty charts to the Barra fishermen, which they were familiar and comfortable with, and would also on occasion correct. Maps here were tools used to gather data on other ways of knowing a landscape. The project noted that whilst the fishermen's knowledge of the sea may not have the perceived objective knowledge of science, it arguably offered a more

---

[17] By making these traditions visible, as in this project, these cases of 'intangible' cultural heritage are made tangible. The MGS became the first UK organisation to become accredited as an expert NGO advisor to UNESCO on ICH. Interview available at: https://ich.unesco.org/en/state/united-kingdom-of-great-britain-and-northern-ireland-GB (accessed on 7 July 2017).

complete understanding, having both practical and emotional power. The fishermen also detailed sustainable practices of fishing, also often embedded in evolving language of place names, for example, to denote that stock was depleted and should be left alone. Critically, the report noted that international treaties specifically requires signatories to 'respect, preserve and maintain knowledge, innovations and practices of indigenous and local communities embodying traditional lifestyles relevant for the conservation and sustainable use of biological diversity' (p. 41).[18] The report noted that since 2006, these indigenous ways of living are protected under international law. However, to date the UK has not signed up to the Convention (UNESCO 2019). This policy is underpinned by a Cultural Ecosystems approach as its explicitly links human cultural diversity as an integral part of other ecosystems. It can be argued that Gaelic culture is a prime example of Intangible Cultural Heritage in the UK. This project thus underlines the potential role and value of mapping in terms of policy-making, underlining the connection between politics and mapping.

Harley and Woodward (1987) had argued that cartographic silence becomes a powerful factor when we attempt to read maps critically. Cartographic silence is particularly important in the context of colonial mapping, especially where 'toponymical silence' enacts and compounds colonial violence. As Harley argues, '[c]onquering states impose a silence on minority or subject population through their manipulation of place-names. Whole strata of ethnic identity are swept from the map in what amounts to acts of cultural genocide' (Harley in, Reddleman 2018, p. 37). However, place-names can be aggressively asserted as well as effaced: e.g., Scottish names erased or Anglicised. By mapping these local names, a reversal of *cartographic deafness*, rather than silence, occurs. By paying attention to the naming through mapping, re-discovering what these words mean in the different historic languages spoken and re-attuning our eyes and ears to the sight and sound of old place names, can we re-read the landscape, informed by the past, in the present, to inform the future.

Chandler and Reid (2018) have argued that the valorisation of indigeneity in western thought is a reflection of the 'ontological turn' in anthropology. They claim that modernist understandings of the world are in crisis today, 'as reflected by the western approaches to indigeneity and in particular to indigenous knowledge'. This crisis, they maintain, 'is being played out in the domain of ontopolitics'—the debate over the politics of being (Chandler and Reid 2018, p. 2).[19] The authors are making a case here for global politics, but *Belonging to the Sea* was initiated to examine the motivation of opposition of local communities who felt disempowered by proposals from the Scottish Government's conservation body, Scottish Natural Heritage, to designate two European marine conservation areas in waters off the island of Barra.[20] The research looked behind the political antagonism to explore the maritime traditions and principles of belief and conduct in this Gaelic community: 'Our research suggests that at the heart of these beliefs and conduct is a particular Gaelic expression of a feeling that is universally potential in human beings: the sense of belonging to a home place, and of responsibility for that place. In the Gaelic context, this feeling—described by the Gaelic scholar John MacInnes as a form of 'emotional energy'—is encapsulated by the not easily translateable word 'dúchas' (in Irish) or 'dùthchas' (in Scottish Gaelic). While 'dúchas/dùthchas' is a word of the land, which finds its etymological root in the Gaelic word 'dú/dùth' which is generally understood to mean 'earth', the research undertaken suggested that the emotional energy of belonging and responsibility the word conveys, extends to the waters around the homeland' (Cape Farewell 2014, p. 9).

---

[18]   The European Habitats Directive (92/43 EEC) was in response to the UN's Convention on Biological Diversity (CBD), endorsed at the UN 'Earth' summit in Rio de Janeiro (1992) and the subsequent endorsement of the 'Malawi principles' at the fifth CBD conference in 2000 in Nairobi, Kenya.

[19]   There are parallels with the UK government and its refusal to sign the Intangible Cultural Heritage (ICH) Convention 2003. Scottish Government urged the UK Government to sign ICG in a motion (S5M-11347) raised in the Scottish Parliament in March 2018. These divergent positions between Holyrood and Westminster are likely to become more explicit when the responsibilities for the environment, farming and fisheries may temporarily be returned to Westminster post Brexit (Sim 2018). The Scottish Parliament: 29 of March 2018: https://www.theyworkforyou.com/sp/?id=2018-03-29.23.0.

[20]   The project also considered anther Gaelic speaking community in Ireland experiencing similar opposition to an Irish Government initiative.

Thus, the fluidity of belonging includes once again a consideration of the littoral lands: a connection to both land and sea in these communities on the periphery of Scotland. *Dùthchas na Mara/Dúchas na Mara/Belonging to the Sea* encompasses a totality of understanding, 'not so much a landscape, not a sense of geography alone, but a formal order of experience in which all these are merged' (MacKinnon and Brennan 2012). Thus, an older and deeper way of knowing the sea, which is distinct from, but potentially complementary to more contemporary ways of knowing, informed by book based learning and formal education. It is noteworthy that there are similarities between these ancient traditions described here, and Shinto in Japan, which in essence assumes a holistic approach to life and nature: *Living as Art Form* as evident in Morton's work and alluded to by Thompson (2012). This was exemplified by fishermen talking of 'going up north'—counter to the cartographic convention—but instead informed by the direction of the prevailing winds or the position of the sun (MacKinnon and Brennan 2012, p. 19). The research also chronicled the speed of change wrought by technology and 'human greed'. There was a real fear among island people that if the ability to fish is removed, then the island's very reason for being a place of human dwelling begins to unravel, surely emblematic of the conundrum of the Anthropocene. *Belonging to the Sea* 'mapped' the intangible cultural heritage of the island, not only by geographically locating knowledge, but by collating it, acknowledging it and making it 'visible. The role of the artist, Stephen Hurrel, was in a sense subservient to the overall aim of the project and he worked predominantly as documenter and recorder, to illustrate points made in the research through his photography. The photographs gave shape and form to this body of intangible traditional knowledge. The conclusion reached by the researchers was that the current lack of connection between these different ways of understanding the world, contributes to the ongoing conflicts on the islands, between (well intentioned) policy and real, lived experience. By *mapping* the indigenous ways of knowing the world, the project has given its community political agency; not as a vindication of resilience, but as a means of securing its future heritage. The case study of *Dùthchas na Mara/Dúchas na Mara/Belonging to the Sea* laid bare the dichotomy between the *lived* practices at micro level, and the policies at macro level, and those caught in between. This collaborative project also gave rise to further collaborations between Hurrel and Brennan.

*Sea Stories* (2013) developed out of *Dùthchas na Mara/Dúchas na Mara/Belonging to the Sea* (2012) and manifested as a digital interactive online map—proposed by Hurrel—and developed by Hurrel and Brennan.[21] The map (Figures 10 and 11) collates the many layers of cultural knowledge of the local waters of Barra. It draws on local knowledge of uncharted coastal and maritime topography around Barra, thus enabling it to be stored, accessed and shared in a visually interesting, interactive format. It was conceived as a means of bringing to life and making visible that which is often invisible: knowledge of local shipwrecks, hidden geographical topographies such as reefs, stories belonging to the shore and the sea, story names of coastal features. This interactive map belongs to the community who make it come alive with their stories, anecdotes and tales, and reflects intergenerational knowledge. It features not only the location and names of shipwrecks and reefs, but also the location of fishing marks that triangulate positions at sea with two landmarks and the 'story names of local features' that detail the geographical information embodied in the Gaelic place names. The map also defines cultural geographies; it captures the stories and anecdotes of the people who have lived and worked in this watery landscape: stories of lives lived and lost at sea (Figure 12). Whilst Hurrel did not report any conflict on what was actually being mapped, he did observe conflict in the community in who 'spoke' for the community (interview 1 February 2019).[22] Knowledge is always contested and as such, the researchers set out to gather diverse contributions.

---

21  http://www.mappingthesea.net/barra/ (accessed on 8 March 2016).
22  This related to whom was chosen to present the project, and where, rather than whom it represented or how it was represented. Ultimately, the community could add to the map themselves without intermediaries from Hurrel or Brennan.

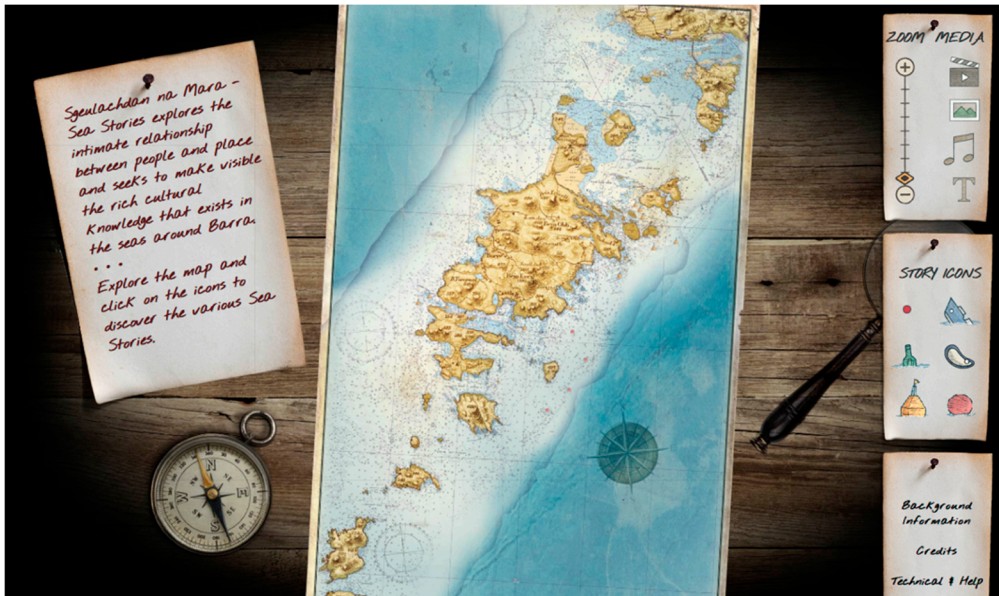

**Figure 10.** Screenshot of Bogha a Chlèirich. Sea Stories website.[23] Image used with permission form the artist.

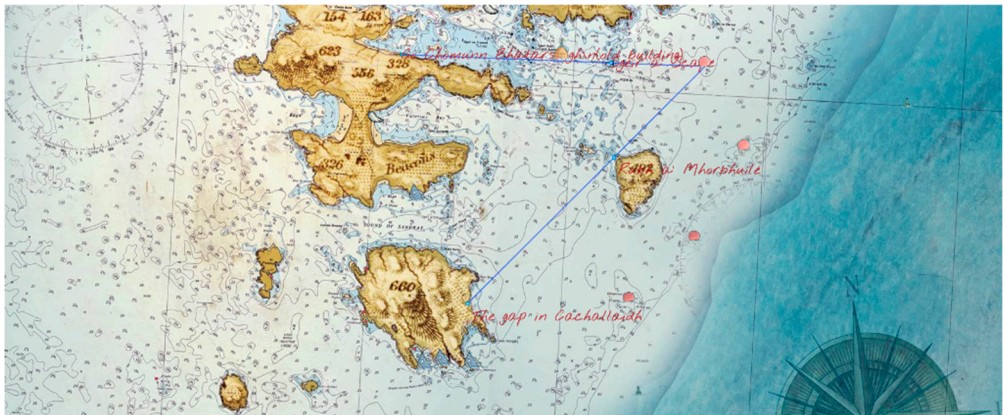

**Figure 11.** Screenshot of the Cullen shipwreck story in Sea Stories website. Reproduced with permission of the artist. Image used with permission form the artist.

Hurrel and Brennan noted that this indigenous knowledge is getting lost, in part through the loss of the Gaelic language but also because GPS has superseded 'the need to name places, to mark them: to make people understand what it is, visualise it [by naming it]' (Hurrel in interview). The fishermen were able to visualise the seabed through these oral traditions long before mapping technologies were able to represent it in three dimensions through sonar mapping. As recalled by Hurrel in an interview, Calum a'Chal made the point: 'But that is not all you need to know. You need to know what the current is like, what the tide is like, what the current might be like at the bottom of the sea, compared to the top of the sea'.

---

23  http://www.mappingthesea.net/barra/.

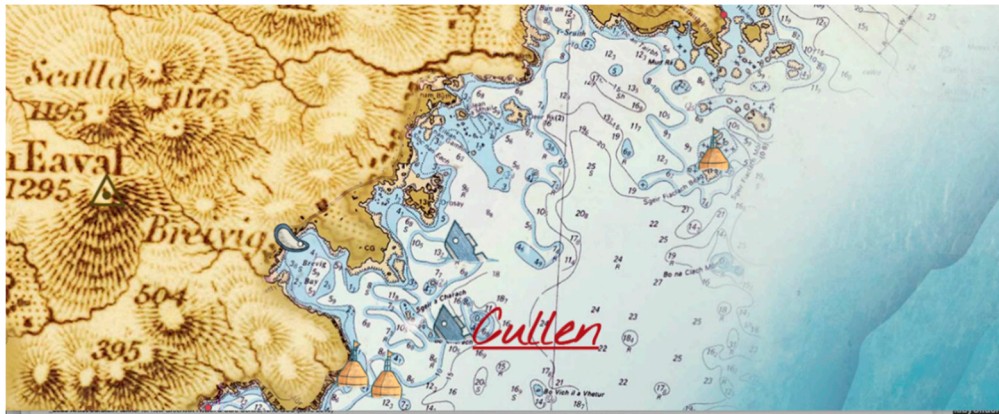

**Figure 12.** Screenshot of the Cullen shipwreck story in Sea Stories website. Reproduced with permission of the artist. Image used with permission form the artist.

*Mapping the Sea: Barra* (2013), as commissioned by Cape Farewell as part of *Sea Change*, was thus the culmination of an extensive period of mapping some of the cultural ecosystems of the Isle of Barra.[24] This nine-minute split screen film interweaves the recitation of place names by fisherman Calum a'Chal, with images from the sea (Figure 13). The film begins with large steel letters placed on the beach and the following quote, before it cuts to Calum reciting from memory, place names from around the Barra coast: 'As technological civilization diminishes the biotic diversity of the earth, language itself is diminished (David Abram, The Spell of the Sensuous)'. The interspersing images show individual fishermen working closely with the sea, along with images of an industrial seafood processing factory, 'suggesting a less direct engagement with the marine environment' (Hurrel 2018).

These mapping projects by Hurrel and Brennan can thus too be viewed as Littoral Art, operating between discourses of science and art, art and policy. Theirs too speaks of a profound enmeshment of cultural ecosystems with other ecologies.

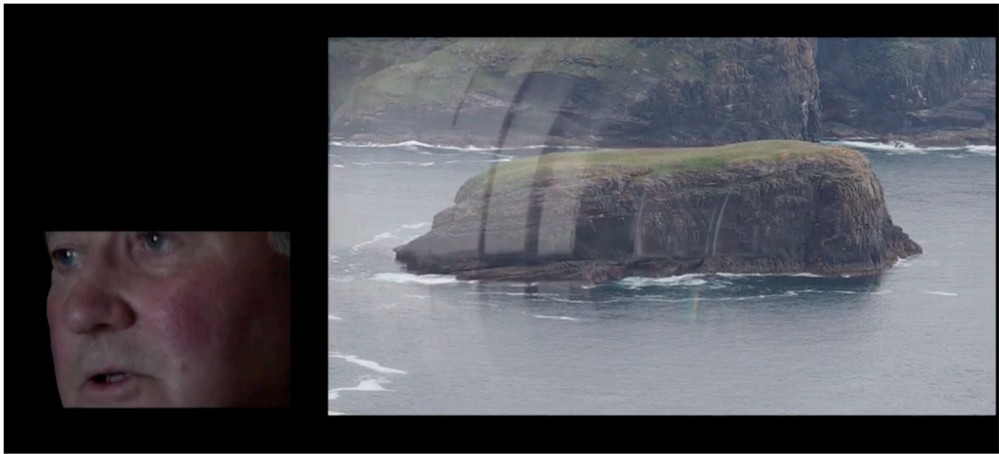

**Figure 13.** Screenshot Mapping the Sea: Barra (2013), Stephen Hurrel, commissioned by Cape Farewell. Image used with permission form the artist.

---

24  *Mapping the Sea: Barra* (2013) formed part of the *North Sea Hitch* (2013) trilogy of works created between 2012 and 2014 and were exhibited at Timespan as one of the most northern outposts of *Generation: 25 years of contemporary art in Scotland* in 2014 The three film pieces were shown consecutively for four weeks; *Dead Reckoning* (2012) 8 June–1 July, *Mapping the Sea: Barra* (2013) 5–29 July and *The Sea, The Sails, and the White, White Blades* (2014) 2–31 August 2014.

## 4. Conclusions

The multiple mapping projects of Barra by Hurrel and Brennan made the embodied knowledge of the sea, of inhabiting the landscape, explicit through the map, in order to be able to demonstrate indigenous knowledge and resilience as examples of Intangible Cultural Heritage to inform conservation policy, whilst Barton's *Neo Terra* mapped the human impact on the local ecology, making explicit the meshwork of material entanglements. *Neo Terra* demonstrated the global macro connections from a micro site (the beaches on the North West coast of Scotland) and of human impact on the environment, demanding policy action through its engagement with policy makers at the Scottish Parliament. Conversely, the Barra projects by Hurrel and Brennan demonstrated how macro events (such as the UN Convention of Biological Diversity) impact upon the micro human environment. Whilst ostensibly, these mapping projects grew out of on the one hand a social science project (*Belonging to the Sea* (2012)), and the other an activism project (*Neo Terra* (2016)), their littoral nature transcend the delineated territories of audience, participant, artist, fisherman, scientist, policymaker etc. The artworks discussed in this paper, were the result of a *process* that engaged with different constituents at different times. The artworks made a difference to the local communities to which they related, by facilitating some political agency, which was perhaps made most explicit in *Belonging to the Sea* (2012) and its engagement with UNESCO's Intangible Cultural Heritage and *Neo Terra* (2016) through its association with the Scottish Parliament. The artworks, although encountered aesthetically as exhibits in art galleries and museums, or outside of it, in community halls, schools and political institutions, are works of signification, which are political and affective. Whilst ostensibly speaking about loss and change, the mapping of the sea through these artworks revealed the *meshwork* of floating, fleeting connections that foregrounds the cultural ecosystems of these communities in these northern peripheries and demonstrates their *enmeshment* with other ecosystems: of sea currents, fish, wind, human activity, global policy making, local government etc. As such, by mapping these cultural ecosystems on the edges of these littoral seascapes, these communities can locate their landscapes, not on perceived peripheries, but enmeshed in larger ecosystems.

**Funding:** AHRC funded PhD research, Northumbria University, Visual Culture.

**Acknowledgments:** I wish to thank the artists for their time and contributions and for the kind use of their work in this article.

**Conflicts of Interest:** The author declares no conflict of interest.

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
