# Peer review of "Mapping the Sea on Scotland’s Peripheries"

_arts, 2019_

Round 1

Reviewer 1 Report

This is an interesting and well written paper that intervenes in current discussions surrounding ‘relational aesthetics’, and ‘socially engaged’ or ‘Littoral’ art as well as supplying case studies that should interest followers of non-representational or more-than-representational geographies as well as anthropologists and heritage researchers. Since littoral art is, almost by definition, interdisciplinary, the paper is of interest to researchers working in a range of disciplines.

The selected art projects are presented in ways that suggest they have avoided some of the pitfalls identified by art critics (referenced in the paper) such as Bishop and Kester, which include the risks of cultural appropriation and of mishandling the politics of recognition in ways that (inadvertently?) segregate and limit populations while valorising de-territorialised artists. As an art practice, mapping seems to have assisted these artists in overcoming some of these problems. Nonetheless, as the authors recognise to some extent, historically, the capacity for mapping to ‘facilitate some political agency’ is complicated. A fact that has been recognised by geographers since J. B. Harley, and explored, in the context of digital mapping technologies, by John Pickles.

Patrick Geddes is referenced as an ancestor figure, although recent work has critically reconsidered some of his many-faceted legacies (see “The Sociological Imagination” (2013). The exhibition Geddes staged (with racial anthropologist H. C. Fleure) as part of the Civic Exhibition in Dublin in 1914 arguably supplies a useful example of the way mapping methodologies, supported by cultural institutions, intervene in political struggles to support governmental agendas. Taking the long view, while the work of Geddes and his circle advanced the careers of ‘experts’, this work sometimes had more mixed outcomes for the ‘folk’ it claimed to empower and ‘improve’. For Geddes and his circle mapping tangible and intangible heritage, farming or fishing practices of local communities (or ‘folk’ in Geddesian terms) was connected to racial science that was very much of its time and deserves careful handling. “Think global, act local” is a slogan whose emancipatory potential depends very much on how agency is distributed between the local and the global – Doreen Massey’s work supplies useful insight on the processes of territorialisation that configure these categories, processes which reproduce and distribute social inequality and environmental degradation.

I would have liked to read more about the political tensions raised by these art-mapping practices which are hinted at in the paper but could be more explicitly foregrounded. For example, disputes over who ‘speaks for’ the community are arguably connected to the use of mapping media and technologies which demand the performance of “speaking for” and from a particular kind of space (p12). Anthropologists have long recognised the politics of recognition inherent in similar methodologies – in which it was only the beneficent anthropologist who could ‘give the community a voice’ and represent ‘the local people’ to external authorities. Interdisciplinary methodologies carry with them a history of paternalism that, paradoxically, has tended to replace and efface indigenous voices, at the same time that it creates funding for expert representatives.

Overall the points I have raised are a matter of emphasis and referencing, and do not require a major overhaul of the text. I enjoyed this paper and think it should be published.  

Author Response

First thank you for your thoughtful and helpful comments on the article.

In response to your comments and your observation on the broader historical references of Mapping, I have tried to address this in the newly added introduction which gives a brief overview of the historical context of mapping and how this relates to mapping practices in art and more specifically the environmental and socially engaged art practices discussed here.  

In relation to Geddes, I fully take on board that 'think global, act local' is indeed only in concept attributed to Geddes as per Walter Stephen (2004). Massey does provide a more nuanced approach on processes of territorialisation so have added her as a reference. I focus on Geddes in relation to his ecological and cultural symbiosis, that finds its roots in Scotland and thus seemed more appropriate to reference than perhaps more contemporary thinking but as you noted with the caveat of it being of its time. I hope I have made clear I simply wish to reference his concept as an outlier, ahead of its time, rather than specifics of his practice. I did have a look at 'Encountering the Sociological Imagination' by John Scott in 'C.Wright Mills and the Sociological Imagination' edited by John Scott and Ann Nilsen (2013) as suggested, but could not find any immediate references to Geddes?

The political tensions raised by the mapping projects were not evident in the interview. I have clarified that the tension referred were in reference to whom was to present the project and where, that was contentious, rather than whom or what was represented in the map as ultimately the community could add to the map themselves without intermediaries from Hurrel or Brennan.

I hope this has clarified the points you raised.

Thank you.

Reviewer 2 Report

This is a timely and important topic. The two artist projects you describe and the contextual/ theoretical support you bring in (the notion of littoral art and meshwork) have good potential. However, I have a few questions and suggestions about strengthening your position and argumentation. 

In the abstract I note you announce you will examine mapping methodologies used by the two artists you examine, with particular reference to concept of Littoral Art as characterised by Kester, and also use of meshworks as a model of enquiry and practice. You close with Patrick Geddes, in fact you highlight Geddes because his 'legacy is relevant to activist mapping discussed here'. I have focused on these key areas in my response, but I also will suggest you improve your introduction, conclusion and some passages that are too compressed.

Mapping methodologies: you certainly discuss practices and completed work that use the appearance and 'tropes' as you have it of mapping, but you do not examine or critique in great detail what exactly  'mapping methodology (ies)' are or might be--especially in relation to process, engagement, and indeed to the exploitation of the 'flux' of the sea you start with. It strikes me that mapping is so closely bound up with colonisation and resource exploitation that some of this literature and challenges through practice ought to be remarked in an essay on the Anthropocene.

Littoral Art--Kester's insightful essay that you cite brings forward much more strongly the problematics of true engagement or the aim of co-production. But your characterisation of Barton's work that is given pole opening position does not bring that out. Reading between the lines I appreciate Barton has indeed engaged with many actors on the shores she has worked in, but these examples have been overshadowed by the 'visitor book comments' in your text that Kester would rightly critique as evidence for engagement. In addition I am a little uneasy about the Littoral/ Literal word play in Barton's title, because Kester of course was not focusing on the shoreline as such but was presenting a more nuanced range of problems and questions about activist work. In short, I feel this theoretical input is asserted rather than used fully as a mode of analysis and argumentation. I have similar questions about 'meshwork'.

Meshwork: lines 151-184. This passage is packed with incomplete nested references (e.g. 'Latour in Chandler, line 152') and feels a little too closely packed with citations and assertions--with less analysis and criticism in between.

Geddes: The way you use Geddes, in a sense co-opt him, makes me feel when reading that this article is perhaps part of a larger Geddes-centred enquiry, as it is not clear why he is a necessary figure in the argument as you construct it. In addition you are a little too hasty to imply several times that Geddes invented the slogan think global, act local, whereas in reality it is a sentiment that is attributed to him in retrospect for example in W. Stephen (2004). Geddes is certainly an important thinker and forerunner in the environmental activist movement, but could be better introduced and placed in this article. --for example what does he have to do with Barton or Hurrell and their ideas--or Kester, come to that?

Overall we can't immediately feel the overall drive of your argument--and what your thesis about mapping methodologies is. Partly this is because you jump straight in to talking about Barton, then Hurrel--where is your framing statement and introduction? Currently the article is ambiguous in tone, and sits between an exhibition review essay and an academic article that includes case studies. It would be useful to clarify this.

Finally, I suggest you get clarification from your editors about the footnotes, which contain largely unreferenced information. 

Author Response

Thank you for thoughtful and considered comments and feedback.

As suggested, I have added an introduction which places the historical developments of mapping (and its historical narrative of resources and colonisation) in relation to mapping in art and particularly in context of environmental and socially engaged art practices. This is further put in context of Cultural Ecosystems Services as a policy framework that outlines the enmeshment of culture and nature. 

I have expanded some of the paragraphs  to explain better the deep engagement of Littoral Art with many actors in the project. I also similarly, expand a little on the meshwork idea. 

In relation to Geddes, I fully take on board that 'think global, act local' is indeed only in concept attributed to Geddes as per Walter Stephen (2004).  I focus on Geddes in relation to his ecological and cultural symbiosis, for its roots in Scotland. I hope I have made clear I simply wish to reference his concept as an outlier, ahead of its time, rather than specifics of his practice. 

I added further references in the footnotes, and bibliography, were this was relevant.

I hope this has clarified the points you raised.

Thank you.

This manuscript is a resubmission of an earlier submission. The following is a list of the peer review reports and author responses from that submission.